# High-intensity UV laser ChIP-seq for the study of protein-DNA interactions in living cells

Arndt Steube[1,2,3], Tino Schenk[3,4,5], Alexander Tretyakov[1] & Hans Peter Saluz[1,3]

Genome-wide mapping of transcription factor binding is generally performed by chemical protein–DNA crosslinking, followed by chromatin immunoprecipitation and deep sequencing (ChIP-seq). Here we present the ChIP-seq technique based on photochemical crosslinking of protein–DNA interactions by high-intensity ultraviolet (UV) laser irradiation in living mammalian cells (UV-ChIP-seq). UV laser irradiation induces an efficient and instant formation of covalent "zero-length" crosslinks exclusively between nucleic acids and proteins that are in immediate contact, thus resulting in a "snapshot" of direct protein–DNA interactions in their natural environment. Here we show that UV-ChIP-seq, applied for genome-wide profiling of the sequence-specific transcriptional repressor B-cell lymphoma 6 (BCL6) in human diffuse large B-cell lymphoma (DLBCL) cells, produces sensitive and precise protein–DNA binding profiles, highly enriched with canonical BCL6 DNA sequence motifs. Using this technique, we also found numerous previously undetectable direct BCL6 binding sites, particularly in condensed, inaccessible areas of chromatin.

[1] Department of Cell and Molecular Biology, Leibniz Institute for Natural Product Research and Infection Biology–Hans Knöll Institute (HKI), Jena, 07745, Germany. [2] Department of Internal Medicine IV, Jena University Hospital, Friedrich Schiller University, Jena, 07747, Germany. [3] Friedrich Schiller University, Jena, 07737, Germany. [4] Department of Hematology and Medical Oncology, Clinic of Internal Medicine II, Jena University Hospital, Jena, 07747, Germany. [5] Institute of Molecular Cell Biology, Center for Molecular Biomedicine Jena (CMB), Jena University Hospital, Jena, 07745, Germany. Correspondence and requests for materials should be addressed to A.S. (email: arndt.steube@med.uni-jena.de) or to T.S. (email: tino.schenk@med.uni-jena.de) or to H.P.S. (email: hanspeter.saluz@leibniz-hki.de)

Genome-wide profiling of protein–DNA interactions is generally performed by chromatin immunoprecipitation in combination with deep sequencing (ChIP-seq)[1–3]. Interacting proteins are chemically crosslinked to their target DNA sequences by formaldehyde (FA), the purified chromatin is sheared and the relevant protein is enriched by immunoprecipitation with specific antibodies. The co-purified genomic DNA is then determined by deep sequencing.

Although conventional ChIP-seq studies have yielded many important insights, limitations and the potential for systematic biases have been identified[4–12]. Formaldehyde crosslinking generates protein–protein and protein–DNA formations, thus disallowing for the discrimination between direct and indirect protein–DNA interactions in subsequent analyses (Fig. 1a)[4]. Protein–protein crosslinking may lead to the identification of artifactual protein–DNA binding, in particular at highly accessible loci[5–7]. Formaldehyde treatment can cause the destruction or masking of epitopes[8,9] and is known to affect the sensitivity of chromatin to fragmentation[10]. In addition, highly dynamic protein–DNA interactions might become undetectable through formaldehyde based ChIP[11,12].

The ChIP technique was introduced by Gilmour and Lis in the 1980s for the detection of direct protein–DNA interactions in vivo[13,14]. The method was originally based on covalent photochemical crosslinking of protein–DNA interactions using germicidal lamps emitting low-intensity ultraviolet (UV) light at relevant wavelengths. UV irradiation results in the formation of covalent "zero-length" crosslinks, which occur exclusively between nucleotide bases and protein amino acids that are in immediate contact (Fig. 1a)[15]. Photochemical crosslinking by low-intensity UV irradiation was used to study several transcription factors at individual loci in *Drosophila* cells[16–18]. However, in mammalian cells, mapping of transcription factors by low-intensity UV crosslinking and subsequent ChIP proved to be inefficient and of low sensitivity[19,20]. Due to the emission of a broad spectrum of UV wavelengths by conventional low-intensity germicidal lamps, a long irradiation time is necessary to obtain sufficient protein–DNA crosslinks, leading to DNA and protein damage[13,14,16]. In contrast, high-intensity UV laser irradiation at 266 nm leads to efficient and virtually instantaneous photochemical crosslinking of protein–DNA interactions in vitro and in vivo[11,15,21–23]. The irradiation time can be significantly shortened, preventing the possibility of artifact formations due to protein redistributions during the crosslinking process and minimizing DNA and protein damage[11,24,25].

In this study we present the first application of photochemical crosslinking by high-intensity nanosecond-pulsed UV laser irradiation in combination with ChIP-seq (UV-ChIP-seq) in living mammalian cells. To evaluate UV-ChIP-seq we investigated genome-wide DNA binding of the sequence-specific transcription factor B-cell lymphoma 6 (BCL6) in human diffuse large B-cell lymphoma (DLBCL) cells[26]. BCL6 is a well-characterized transcriptional repressor playing important roles in the formation of germinal centers (GC) during immune responses and in the initiation and maintenance of B-cell lymphomas[27]. The genome-wide binding of BCL6 has been extensively studied using conventional FA ChIP techniques, revealing thousands of potential BCL6 binding sites[27–32]. Nevertheless, most sites found did not overlap canonical BCL6 DNA sequence motifs. In contrast, the UV-ChIP-seq technique presented here results in the detection of robust and high quality genome-wide BCL6-DNA binding sites with high specificity and resolution. Our technique

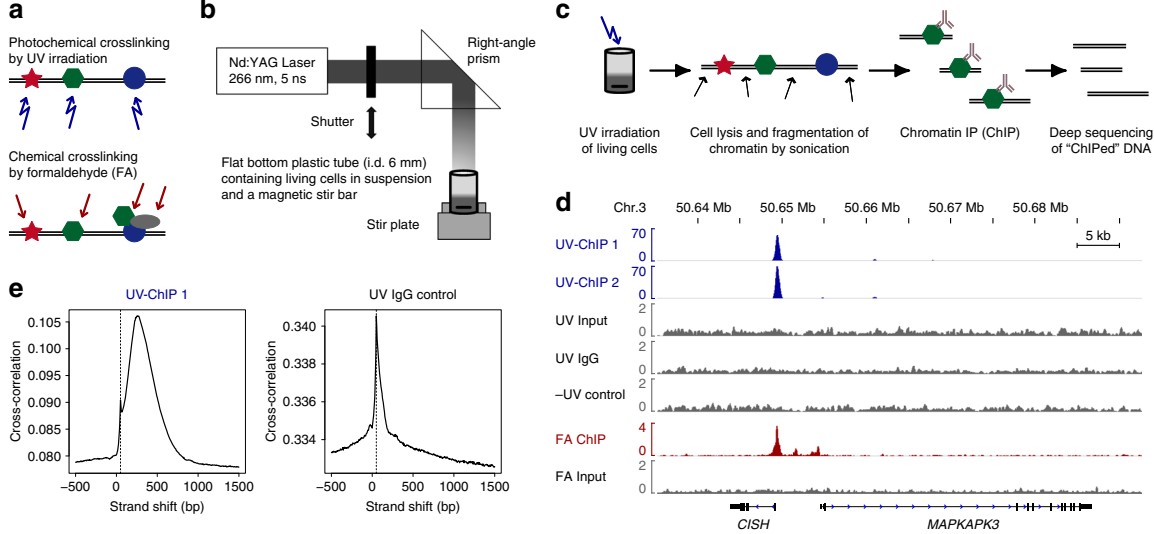

**Fig. 1** High-intensity UV-ChIP-seq for the study of BCL6-DNA interactions. **a** Crosslinking strategies. Photochemical crosslinking by UV irradiation results in the formation of covalent "zero-length" protein–DNA crosslinks. Chemical crosslinking by formaldehyde (FA) fixates protein-protein and protein–DNA interactions via methylene bridges. **b** Experimental setup for high-intensity UV laser irradiation of living cells. A pulsed laser beam of 266 nm was focused and adjusted to fit the surface of the sample area. Cells were irradiated at 4 °C under constant stirring. **c** UV-ChIP-seq workflow. Following UV laser irradiation, cells were lysed and chromatin was fragmented by sonication. Subsequently, specific protein–DNA complexes were enriched by chromatin immunoprecipitation, the co-purified DNA ("ChIPed" DNA) was isolated and analyzed by deep sequencing. **d** ChIP-seq profiles of an example region containing a validated BCL6 binding site. Read density profiles of biological replicate BCL6 UV-ChIP-seq (UV-ChIP 1 and 2, blue tracks) and UV control (UV input DNA, UV IgG control and -UV control ChIP, gray tracks) data are shown for the human *CISH, MAPKAPK3* loci. UV-ChIP-seq detects specific enrichment within the promoter region of the *CISH* gene, overlapping a DNA sequence previously found by FA ChIP-seq, containing a canonical BCL6 binding motif (red track). UV control profiles show no enrichment. Note the different scales in ChIP and control profiles. ChIP-seq read density profiles are shown as pileup signal track. Genomic coordinates and human RefSeq annotations (GRCh37/hg19) are indicated. **e** Strand cross-correlation (SCC) analysis of UV-ChIP-seq data. SSC analysis showed a high level of read clustering. Plot profiles are shown for UV-ChIP-seq replicate 1 (UV-ChIP 1) and UV IgG control data. Dashed lines indicate the strand shifts corresponding to the sequenced read length (50 bp) (Supplementary Fig. 4)

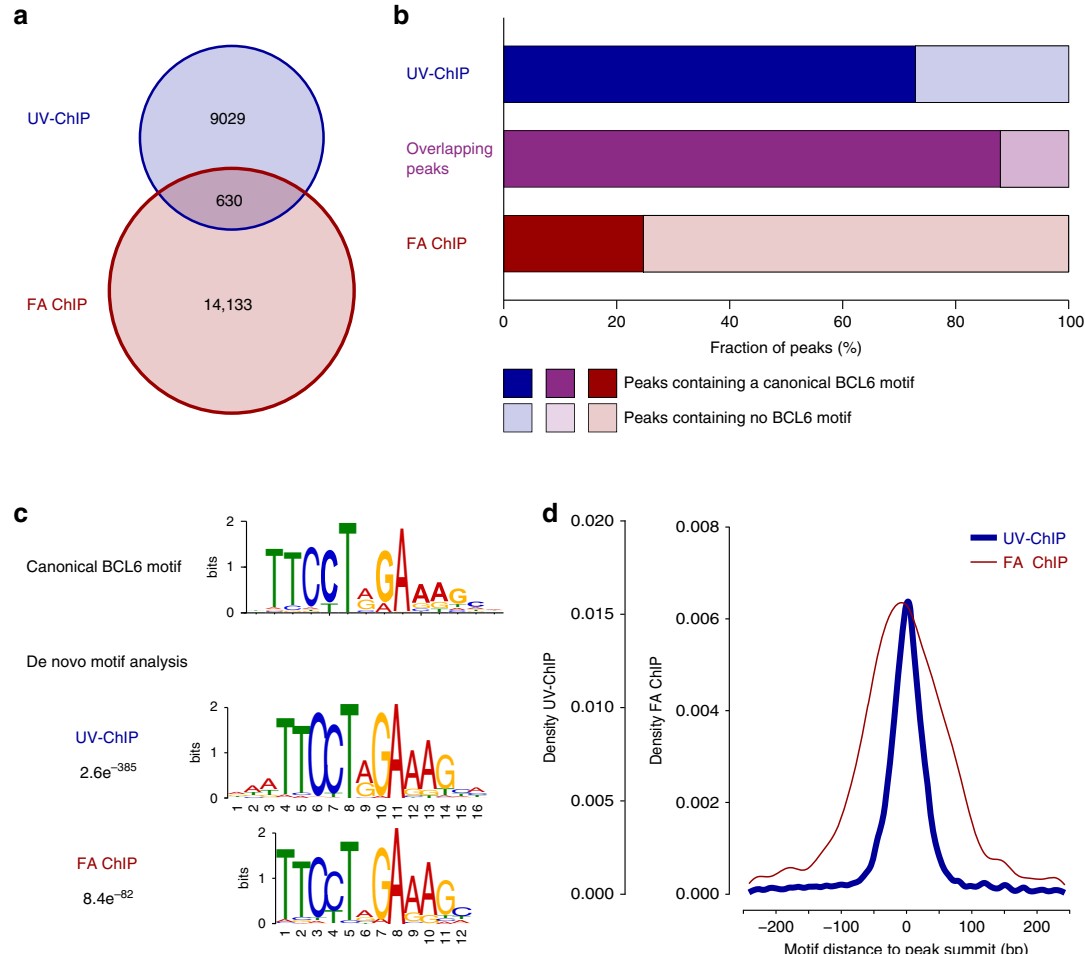

**Fig. 2** Binding sites identified by UV-ChIP-seq are enriched with canonical BCL6 motifs. **a** Detected BCL6 binding sites (peaks) in human DLBCL cells. The number of BCL6 peaks detected by both UV-ChIP-seq ($n = 9029$; blue), FA ChIP-seq ($n = 14,133$; red) and the corresponding peak overlap ($n = 630$) is shown (see also Supplementary Fig. 9). **b** Occurrence of canonical BCL6 motifs within detected peaks. The fraction of peaks (%) containing a canonical BCL6 motif ($q$-value 0.1, dark colored) and the fraction that does not contain a BCL6 binding motif (light-colored) are plotted for UV-ChIP-seq (72.9% peaks with motif, blue), FA ChIP-seq (24.7%, red) and overlapping binding sites (88.0%, purple). **c** DNA sequence motif analysis. The canonical BCL6 motif and the corresponding de novo DNA sequence motifs discovered from UV- and FA ChIP-seq data analyses are shown and $E$-values are indicated (Supplementary Fig. 7). **d** Distance of BCL6 motifs relative to peak summits. The distance of canonical BCL6 motifs is plotted relative to the corresponding peak summit (base position of maximum enrichment, $x$ axis, $\pm 250$ bp) as detected by UV-ChIP-seq (median $\pm 18$ bp, blue) and FA ChIP-seq (median $\pm 58$ bp, red)

enables the accurate and precise discovery of many previously undetectable direct BCL6 binding sites, particularly in condensed, inaccessible areas of chromatin.

## Results

**UV-ChIP-seq of BCL6-DNA interactions**. For photochemical crosslinking of protein–DNA interactions we irradiated human DLBCL cells using a high-intensity nanosecond-pulsed UV laser technique. The experimental setup for UV laser irradiation of living cells and the UV-ChIP-seq workflow is shown in Fig. 1b and c. In brief, a UV laser beam of 266 nm was generated by quadrupling the main frequency of a Nd:YAG laser (1064 nm), focused and adjusted to fit the surface of the sample area. Precooled cells in suspension were irradiated under constant stirring at 4 °C, enabling uniform irradiation of all cells. Following irradiation, cells were lysed using non-denaturing buffers and chromatin was fragmented by mild sonication. The final step saw specific protein–DNA complexes being enriched by chromatin immunoprecipitation (ChIP) and the co-purified DNA was isolated and analyzed by deep sequencing (UV-ChIP-seq).

Photochemical crosslinking efficiency and consequently nucleoprotein enrichment depends in great parts on the total UV dose applied. Therefore, we first identified the optimum UV dose obtaining maximum enrichment of crosslinked BCL6-DNA complexes in DLBCL cells by UV-ChIP-qPCR of known BCL6 binding sites within the promoter regions of the *CISH*, *AFF3* and *FPGT* genes. In brief, cells were irradiated by increasing doses of UV light (3.5–35 J cm$^{-2}$), lysed and chromatin was fragmented by sonication. DNA bound by BCL6 was immunoprecipitated using an anti-Bcl6 antibody, isolated and analyzed by qPCR—amplifying previously validated areas containing canonical BCL6 binding motifs (Supplementary Fig. 1 and Supplementary Table 1). Our experiments revealed an optimum UV dose of 8.8 J cm$^{-2}$ using the UV laser technique described in the Methods. This UV dose showed minimal effects on the amplificability of DNA fragments of different length as measured by qPCR (Supplementary Fig. 2).

For UV-ChIP-seq experiments, DLBCL cells were irradiated and immunoprecipitation of BCL6 was carried out as described, followed by deep sequencing of enriched DNA fragments. To control for enrichment of non-crosslinked protein–DNA

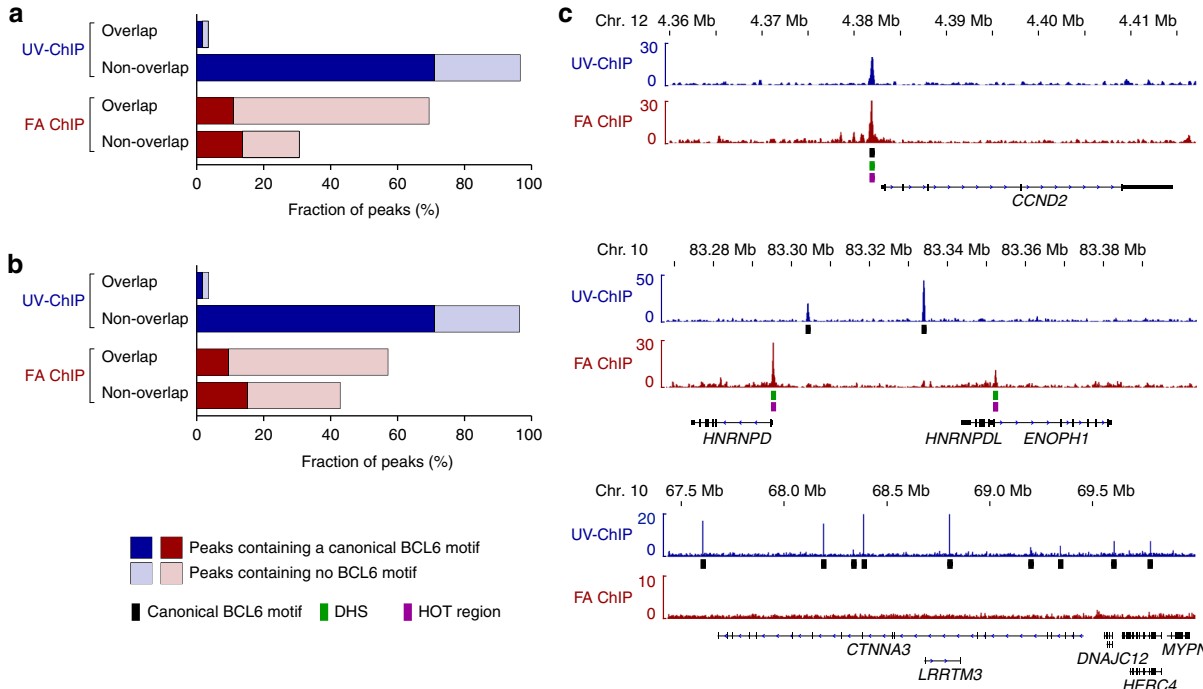

**Fig. 3** UV-ChIP-seq reveals preferential BCL6 binding in less accessible chromatin regions. The fraction of BCL6 peaks (%) overlapping or non-overlapping. **a** DNaseI Hypersensitive Sites (DHSs) in human DLBCL cells and **b** Highly Occupied Target (HOT) regions in human B cells detected by UV-ChIP-seq (blue) and FA ChIP-seq (red) are shown. The proportion of peaks containing a canonical BCL6 motif (q-value 0.1) is indicated (dark-colored). **c** Examples of BCL6 ChIP-seq binding profiles at different genomic loci. (Upper Panel) BCL6 binding within the promoter region of the known target gene *CCND2*. UV-ChIP-seq (blue track) reveals specific enrichment within an accessible chromatin region (DHS/HOT region) in the *CCND2* promoter region overlapping a canonical BCL6 motif. This binding site was also detected by FA ChIP-seq (red track). (Middle Panel) Example of differential BCL6 binding. BCL6 binding profiles show enrichment at different locations within the *HNRNPD* human genomic loci. BCL6 binding sites detected by UV-ChIP-seq overlap canonical BCL6 motifs. FA ChIP-seq identified BCL6 binding within DHSs and HOT regions to non-canonical DNA sequence motifs. (Lower Panel) Profiling of BCL6 binding within a region of late replicating heterochromatin. UV-ChIP-seq detected several binding sites containing canonical BCL6 motifs within the heterochromatic region at chr.10q21.3, which were not identified by FA ChIP-seq. Read density profiles are shown as fold enrichment track generated by ChIP over input DNA pileup signal with the corresponding scales as indicated. Genomic coordinates and human RefSeq annotations (GRCh37/hg19) are shown. Black boxes indicate canonical BCL6 motif appearance (q-value 0.1). Peaks overlapping DHSs (green boxes) and HOT regions (purple boxes) are marked

interactions we performed ChIP-seq using non-irradiated cells (−UV control ChIP). This control showed no enrichment indicating that non-covalent BCL6-DNA interactions are disrupted by the wash conditions (Fig. 1d and Supplementary Fig. 3). Furthermore, unspecific antibody binding and input loading was controlled by sequencing of IgG enriched DNA fragments (UV IgG control) and input DNA fragments (UV input DNA) following UV irradiation, respectively. The obtained sequence data are summarized in Supplementary Table 2.

**Genome-wide mapping of BCL6 binding sites**. Deep sequencing of UV-ChIP derived DNA fragments revealed well defined read enrichments relative to UV control experiments (Fig. 1d and Supplementary Fig. 3). We performed strand cross-correlation (SCC) analysis to assess signal-to-noise ratios following guidelines developed by the ENCODE consortium[33,34]. UV-ChIP sequence data exhibited a high level of read clustering (RSC > 2, QC tag 2) whereas the UV control experiments (UV input DNA, UV IgG control and −UV control ChIP) showed no enrichment (RSC < 1, QC tag ≤ 0) (Fig. 1e, Supplementary Fig. 4 and Supplementary Table 2).

To detect BCL6 binding sites (peaks) within the obtained UV-ChIP-seq data we first performed peak calling using the Irreproducible Discovery Rate (IDR) framework[34]. This algorithm detected 6910 BCL6 binding sites (IDR 0.01) within UV-ChIP-seq biological replicate data sets relative to UV input

DNA data (Supplementary Fig. 5 and Supplementary Table 3). Similar results were obtained relative to UV IgG control data as control for background enrichment (Supplementary Table 3). To test the reproducibility and quality of individual experiments we performed IDR analysis on UV-ChIP-seq replicate (Nt) and pseudo-replicate (Np, pooled and subsampled reads from replicates) data sets. This analysis revealed a high reproducibility (Np/Nt = 1.04) and confirmed the high quality of each individual experiment (N1/N2 = 1.11) (Supplementary Fig. 6 and Supplementary Table 3). In addition to the IDR analysis, we performed peak detection of pooled sequence data using the MACS algorithm[35]. Via this approach, we identified 9029 BCL6 binding sites (q-value $1e^{-2}$) in human DLBCL cells.

**Binding sites are enriched with canonical BCL6 motifs**. De novo motif analysis within DNA sequences detected by UV-ChIP-seq revealed strong overrepresentation (E-value $2.6e^{-385}$) of a DNA sequence matching the canonical BCL6 binding motif previously reported (Fig. 2b and Supplementary Fig. 7a)[36]. A position weight matrix (PWM) scan within DNA sequences for the canonical BCL6 motif showed a strong enrichment in the majority of identified binding sites (72.9%, n = 6580, q-value 0.1) (Fig. 2c). In addition, canonical BCL6 motifs strongly clustered (median ± 18 bp) around the peak summit (base position of maximum enrichment) within the detected binding sites (Fig. 2d).

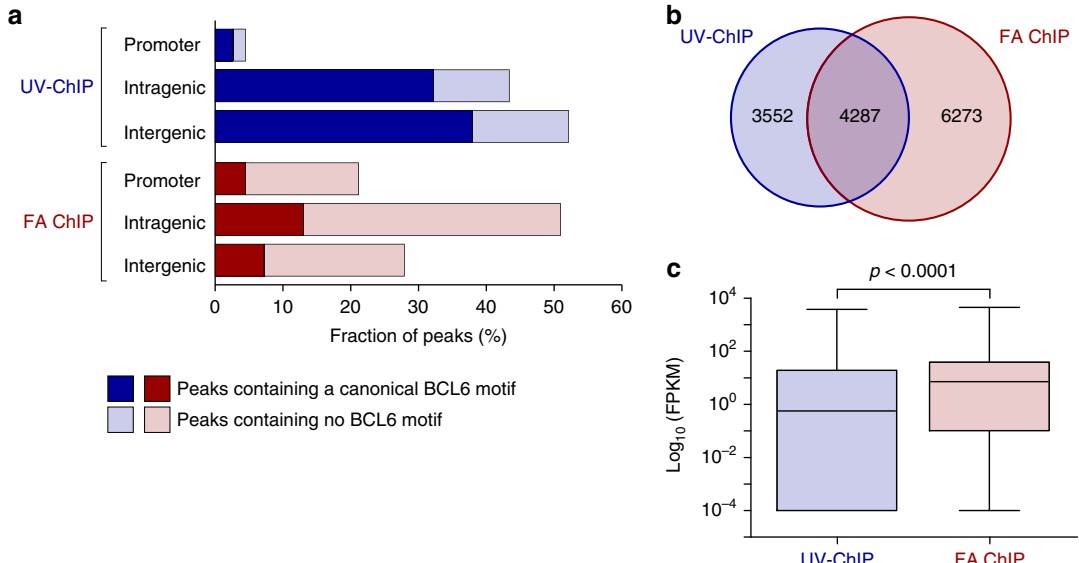

**Fig. 4** UV-ChIP-seq reveals preferential BCL6 binding to transcriptionally less active regions in human DLBCL cells. **a** Location of BCL6 binding sites relative to protein coding regions. The fraction of BCL6 binding sites (%) at promoter regions of annotated genes, intragenic (intron, exon, UTRs) and intergenic regions is shown as analyzed by UV-ChIP-seq (blue) and FA ChIP-seq (red) analyses. The proportion of peaks containing a canonical BCL6 motif ($q$-value 0.1) is indicated (dark-colored). **b** Target gene prediction based on identified BCL6 binding sites. The Venn diagram shows the number and overlap of potential BCL6 target genes associated with UV-ChIP-seq ($n = 7839$; blue) and FA ChIP-seq ($n = 10,560$; red) detected binding sites. **c** Expression status of potential BCL6 target genes identified by RNA-seq in DLBCL cells. Gene expression values (Log10 FPKM, fragments per kilobase of transcript per million mapped reads) of UV-ChIP-seq (blue) and FA ChIP-seq (red) predicted BCL6 target genes are shown by box plots (black line represent the median and whiskers the min. and max. value)

**BCL6 binding is found in inaccessible chromatin regions**. Most transcription factors have been found to preferentially bind within accessible chromatin regions[33,37]. In order to determine whether BCL6 binding occurs preferentially in accessible or inaccessible chromatin regions, binding sites were correlated to DNaseI hypersensitive sites (DHSs) in human B-cell lymphoma cells. We found that UV-ChIP-seq detected the majority of BCL6 binding sites in DNaseI insensitive regions (96.6% of peaks non-overlap DHSs, $n = 8723$) (Fig. 3a). Interestingly, correlation of binding sites to the DHS master list from ENCODE (DHSs detected in 125 cell types) revealed 53.1% ($n = 4794$) of identified BCL6 peaks did not overlap a known DHS (Supplementary Fig. 8)[37]. To deepen these analyses, we examined BCL6 binding within Highly Occupied Target (HOT) regions in human B cells[38]. These regions of highly accessible chromatin are believed to be dynamically bound by a great variety of transcription factors and often do not contain specific DNA sequence motifs[38]. UV-ChIP-seq showed minimal binding of BCL6 (3.6%, $n = 322$) in these parts of the genome (Fig. 3b). Analyses of the genomic location of identified BCL6 binding sites relative to protein coding regions revealed that 4.5% ($n = 405$) were located in promoter regions of annotated genes, 43.4% ($n = 3918$) in intragenic, and 52.1% ($n = 4706$) in intergenic regions (Fig. 4a).

**Comparative analysis of BCL6 ChIP-seq data**. To compare photochemical (UV) with conventional chemical (FA) cross-linking, we performed BCL6 FA ChIP-seq. Except for the method used for protein–DNA crosslinking, cell lysis and sonication conditions, FA ChIP-seq was performed identically using the same human DLBCL cell line (OCI-LY1) and antibody type (Bcl6 (N-3)) as used for the UV-ChIP-seq experiments. In addition, we collected FA ChIP-seq data from two independent studies elucidating BCL6 binding in OCI-LY1 cells (GSE29282 and GSE46663)[30,31]. The raw sequence data was analyzed identically using the mapping and peak calling procedures described above.

Following peak calling, BCL6 FA ChIP-seq data sets have been merged for comparative analysis to UV-ChIP-seq data. This analysis revealed 14,133 BCL6 binding sites detected by all three FA ChIP-seq experiments in comparison to the 9029 binding sites identified by UV-ChIP-seq (Fig. 2a). Thereby 630 BCL6 binding sites were detected by both, UV and FA ChIP-seq, techniques (Fig. 2a and Supplementary Fig. 9).

Motif analysis in FA ChIP-seq identified binding sequences showed a lower percentage of canonical BCL6 motifs (24.7%, $n = 3497$, $q$-value 0.1) when compared to UV-ChIP-seq data (72.9%, $n = 6580$, $q$-value 0.1) (Fig. 2b). Consequently, de novo motif analysis found the BCL6 motif being overrepresented but at a lower alignment score ($E$-value $8.4e^{-82}$) (Fig. 2c and Supplementary Fig. 7b). Canonical BCL6 motifs within DNA sequences obtained by FA ChIP-seq occurred further away from the peak summit (median $\pm 58$ bp) as compared to UV-ChIP-seq, which detected direct BCL6 interactions with improved resolution (~3-fold) (Fig. 2d). Remarkably, almost all BCL6 binding sites which were identified by both, UV-ChIP-seq and FA ChIP-seq contained the canonical BCL6 binding motif (88.0%, $n = 554$) (Fig. 2b and Supplementary Fig. 9). In contrast to UV-ChIP-seq, which detected BCL6 binding predominantly in inaccessible chromatin regions (overlap DHSs and HOT regions < 3.6%), binding sites found by FA ChIP-seq overlapped to a great extent with DHSs (69.4%, $n = 9810$) and HOT regions (57.1%, $n = 8076$) (Fig. 3a, b and Supplementary Fig. 8). FA ChIP-seq detected 21.2% ($n = 2990$) of BCL6 binding sites within promoter regions, 50.9% ($n = 7199$) in intragenic, and 27.9% ($n = 3944$) in intergenic regions (Fig. 4a). In comparison to UV-ChIP-seq data analyses, FA ChIP-seq revealed a significant percentage of BCL6 binding sites within promoter regions of annotated genes, mostly to non-canonical DNA sequence motifs (Fig. 4a).

**UV-ChIP-seq detects known and new BCL6 binding sites**. Among the BCL6 binding sites identified by UV-ChIP-seq were

many promoters of known, previously validated BCL6 target genes. For example, BCL6 binding was detected to canonical BCL6 motifs within the *CISH*, *CCND2*, *SOCS2*, and *HELB* promoter regions (Figs. 1d, 3c (upper panel) and Supplementary Fig. 3)[27–29,39,40]. Other BCL6 binding sites did not colocalize with sites found by conventional FA ChIP-seq (Fig. 3c (middle panel) and Supplementary Fig. 10). The vast majority of newly identified BCL6 binding sites using UV-ChIP-seq were found within DNaseI insensitive areas of the DLBCL genome. For example, within the region of constant late replication at chromosome 10 (chr.10q21.3), UV-ChIP-seq detected nine binding sites containing canonical BCL6 DNA sequence motifs, which were previously undetectable (Fig. 3c (lower panel) and Supplementary Fig. 11)[41]. All together UV-ChIP-seq identified 6026 new sequence-specific BCL6 binding sites within the genome of human DLBCL cells, mostly located within condensed chromatin and previously undetectable by conventional FA ChIP-seq. Furthermore, examination of the evolutionary conservation of UV-ChIP-seq detected binding sites among species showed that BCL6 bound DNA sequences were significantly conserved as compared to adjacent control regions, suggestive for their functional relevance (Supplementary Fig. 12).

**BCL6 target genes show low transcriptional activity**. Target gene prediction based on identified BCL6 binding sites in human DLBCL cells revealed 7839 potential target genes associated with UV-ChIP-seq and 10,560 in the vicinity of FA ChIP-seq peaks (Supplementary Data 1)[42]. A number of 4287 genes were predicted by both, UV and FA ChIP-seq data analyses independent of a corresponding peak overlap (Fig. 4b). To evaluate the transcriptional activity of the potential BCL6 target genes we analyzed their expression level by RNA-seq in OCI-LY1 cells. This analysis revealed that genes associated with UV-ChIP-seq identified BCL6 binding sites showed significantly lower transcriptional activity when compared to the FA ChIP-seq target gene analysis (Fig. 4c). These findings support the observation that less than two percent of UV-ChIP-seq identified BCL6 binding sites overlap with chromatin areas harboring transcription promoting histone modifications such as acetylated lysine 27 of histone 3 (H3K27ac, peak overlap < 1.7%) and trimethylated lysine 4 of histone 3 (H3K4me3, peak overlap < 2.0%) in OCI-LY1 cells (Supplementary Fig. 13). Differential gene expression analyses following knockdown of BCL6 in OCI-LY1 cells showed the significant differential regulation of 747 genes (*q*-value 0.05) associated with BCL6 binding sites identified by UV-ChIP-seq (Supplementary Fig. 14, Supplementary Data 2)[30]. A number of 185 genes thereof were not predicted by FA ChIP-seq analyses and are likely to represent new functional BCL6 target genes in DLBCL cells (Supplementary Data 2).

## Discussion

Genome-wide profiling of protein–DNA interactions is customarily performed by ChIP-seq[1–3]. In this manuscript we describe the first application of the ChIP-seq technique based on a high-intensity nanosecond-pulsed UV laser setup for the generation of photochemical protein–DNA crosslinks in living mammalian cells. UV laser irradiation induces an efficient and instant formation of covalent crosslinks sufficient to identify the bound DNA by immunoprecipitation and genome-wide sequencing. The optimum UV dose and UV light intensity used for protein–DNA crosslinking in living cells does not result in protein or DNA damage. However, UV laser irradiation induces single and double-strand DNA breaks which enhance the ultrasonic shearing of the crosslinkend chromatin[43].

To validate our method, we investigated the genome-wide DNA binding of the sequence-specific transcriptional repressor BCL6 in human DLBCL cells. We found that UV laser irradiated cells can easily be lysed using non-denaturing buffers and their chromatin is uniformly sheared. The use of non-denaturing buffers has the advantage of maintaining epitopes in their native conformation thus potentially opening the path to use non-ChIP-grade antibodies for ChIP. Sequence data obtained by UV-ChIP-seq were reproducible, showed high signal-to-noise ratios and fulfilled ENCODE standards[33]. The specificity and resolution of the detected binding profiles was superior when compared to conventional FA ChIP-seq. The majority of identified binding sites contained canonical BCL6 DNA sequence motifs which were located closer to their corresponding peak summits, thus indicating sequence-specific BCL6-DNA binding. In addition, previously found direct BCL6-DNA interaction sites in DLBCL cells were being corroborated and DNA sequences bound by BCL6 were significantly conserved among species. All aforementioned analyses support the authenticity and functional relevance of the identified BCL6-binding sites.

In contrast to earlier findings, UV-ChIP-seq revealed predominant binding of BCL6 to intra- and intergenic genomic regions and mostly to inaccessible chromatin. In addition, almost none of the identified binding sites were found in chromatin areas with high levels of activating histone marks and genes associated to BCL6 binding sites had low transcriptional activity. As BCL6 is known to suppress transcription by recruitment of histone deacetylases, these results were expected but previously not detected in analyses using conventional FA ChIP-seq.

Studies have indicated that beside its function as a sequence-specific transcriptional repressor, BCL6 contributes to nuclear organization, replication, and chromatin-mediated regulation[44,45]. BCL6 binding identified by UV-ChIP-seq within inaccessible chromatin regions may therefore have architectural functions. This hypothesis offers opportunities for investigation in future studies.

In summary, UV-ChIP-seq results in a "snapshot" of specific nucleoprotein formations by photochemical crosslinking. Our technique produces highly resolved direct protein–DNA binding profiles, strongly enriched with canonical DNA sequence motifs in both condensed and open chromatin regions. Being independent from chromatin accessibility, UV-ChIP-seq is capable to identify previously undetectable direct protein–DNA interactions.

## Methods

**Cell culture**. The B-cell lymphoma cell line OCI-LY1 (GCB-DLBCL OCI-LY1, ACC 722) was maintained in Iscove's Modified Dulbecco's Medium (Gibco) supplemented with fetal calf serum (10% v/v, Gibco) at 37 °C and 5% $CO_2$.

**UV laser technique**. UV laser irradiation of living cells was performed using a Quanta-Ray pulsed Nd:YAG laser (Model GCR-150, Spectra Physics) equipped with an HG-2 harmonic generator (Spectra Physics) and dichroic mirrors (DHS-2 Quanta-Ray dichroic harmonic separator) to give monochromatic light at 266 nm. The laser beam was focused by a fused silica lens, deviated by 90° and adjusted to fit the surface of the sample area. The laser energy at the sample position was determined using a Power/Energy Meter (Nova, Ophir Optronics Ltd.) equipped with a Power Thermal Sensor (Model 10A-P-SH, Ophir Optronics Ltd.). The parameters for UV laser irradiation were as follows: pulse duration 5 ns, repetition rate 10 Hz, energy per pulse 50 mJ, diameter of the laser beam 6 mm.

**UV irradiation of living cells**. For UV-ChIP experiments, OCI-LY1 cells were counted, collected, and washed twice in ice cold PBS (pH 7.4). $1 \times 10^8$ cells were resuspended in 1 ml ice-cold PBS and divided into four aliquots. An open-top, flat-bottom polyethylene sample tube (1 ml, i.d. 6 mm), containing approximately $2.5 \times 10^7$ cells in suspension (250 µl) and a magnetic stir bar, was vertically aligned with the laser beam and positioned on a stir plate. UV laser irradiation was performed at 4 °C under constant stirring (Ikamag RCT, 100 r.p.m.).

**Chromatin immunoprecipitation.** Immediately after irradiation, four aliquots of OCI-LY1 cell suspensions were combined and added to 10 ml of ice cold PBS (pH 7.4) containing protease inhibitors according to the manufacturer's recommendations (Complete Protease Inhibitor, Roche Applied Science). All subsequent steps were performed at 4 °C. After centrifugation at $1000 \times g$ for 7 min, the pellet was resuspended in 250 µl ChIP lysis buffer (0.5% NP-40, 150 mM NaCl, 50 mM Tris-HCl pH 8.0, 5 mM EDTA, protease inhibitors). Chromatin was sheared on ice (Microtip sonicator, Labsonic U, Sartorius) into fragments averaging 250 bp and centrifuged at $16,000 \times g$ for 10 min. Fragmentation of chromatin was analyzed with purified DNA using agarose gel electrophoresis and a 2100 Bioanalyzer (Agilent Technologies). An aliquot of sheared chromatin was used for UV input DNA sequencing and UV-ChIP-qPCR analysis. The supernatant was pre-cleared for 1 h using pre-washed protein A magnetic beads according to the manufacturer's instructions (Dynabeads Protein A, Invitrogen). A ChIP-grade anti-Bcl6 antibody (Bcl-6 (N-3), sc-858, Lot# A3013, Santa Cruz) was used for ChIP. The specificity and suitability for ChIP was confirmed by immunoblot analysis. For UV IgG control ChIP, normal rabbit IgG (sc-2027, Lot# D1513, Santa Cruz) was used. Clarified chromatin extracts were incubated for 12 h with 10 µg of specific anti-Bcl6 antibody and control IgG antibody, respectively. Nucleoprotein-antibody complexes were precipitated using pre-washed beads for 1 h. The supernatant was removed and beads were sequentially washed twice in low salt (0.1% SDS, 1% Triton-X 100, 150 mM NaCl, 20 mM Tris pH 8.0, 2 mM EDTA), high salt (0.1% SDS, 1% Triton-X 100, 500 mM NaCl, 20 mM Tris-HCl pH 8.0, 2 mM EDTA), LiCl (0.5% NP-40, 0.5% deoxycholic acid (DOC), 250 mM LiCl, 10 mM Tris-HCl pH 8.0, 1 mM EDTA) and TE (10 mM Tris pH 8.0, 1 mM EDTA) buffer. For elution of immunoprecipitated nucleoprotein-antibody complexes, beads were resuspended in 0.1 M citrate buffer (pH 2.2) for 2 min at room temperature. The supernatant was removed and transferred into Tris-HCl (pH 8.0) for neutralization. After RNase A treatment (50 µg/ml, Fermentas) for 1 h at 37 °C and Proteinase K treatment (100 µg/ml, Fermentas) for 12 h at 50 °C, genomic DNA was purified by phenol/chloroform/isoamylalcohol extraction and ethanol precipitation.

Formaldehyde (FA) ChIP was performed as described above except for the following steps. OCI-LY1 cells were fixed with 1% FA for 10 min under rotation at room temperature and the reaction was quenched by addition of 125 mM glycine for 5 min. Cells were collected, washed twice with ice-cold PBS (pH 7.4) and lysed in RIPA buffer (1% NP-40, 0.5% DOC, 0.1% SDS, 150 mM NaCl, 50 mM Tris pH 8.0, 5 mM EDTA, protease inhibitors). For elution of immunocomplexes, beads were resuspended in elution buffer (1% SDS, 0.1 M NaHCO₃) and crosslinking was reverted by addition of 0.3 M NaCl and incubation for 5 h at 65 °C.

**UV dose titration for photochemical crosslinking of BCL6.** UV dose titration was performed by UV-ChIP and quantitative real-time PCR (UV-ChIP-qPCR). UV dose (total energy density) for photochemical crosslinking of BCL6 to DNA ranged 3.5–35.3 J cm$^{-2}$. Irradiation of cells and subsequent ChIP was performed as described. DNA fragments enriched by UV-ChIP were analyzed for confirmed sequence-specific BCL6 binding sites and control sites using SYBR Green chemistry (Fast SYBR Green Master Mix, Applied Biosystems) and the StepOnePlus Real-Time PCR System (Applied Biosystems) as recommended by the manufacturer. Variations across samples were adjusted by normalization to input DNA. The $\Delta$Ct values were calculated for anti-Bcl6 and IgG control immunoprecipitated DNA fragments relative to input DNA by $\Delta$Ct [normalized ChIP] = (Ct [Input]−Log2 (100))−Ct [ChIP]. The percentage of input (% Input) values were calculated by % Input = $2^{\Delta\text{Ct [normalized ChIP]}}$. The % Input value represents the enrichment of BCL6 on specific genomic regions. The enrichment of BCL6 relative to control regions and the UV IgG control enrichment at specific sites was calculated.

Target primer pairs span BCL6 binding sites and corresponding control primer pairs (specific control region) are located 2.4–3.2 kb upstream the BCL6 interaction site[27,29,40]. The effect of UV laser irradiation on amplificability of DNA fragments was estimated by high-intensity UV laser irradiation of OCI-LY1 cells with increasing UV doses. Cells were irradiated (energy density 0 to 106 J cm$^{-2}$), total DNA was isolated and qPCR analysis of genomic regions A (295 bp) and B (124 bp) was performed as described. Ct values were normalized to input DNA (non-irradiated control, −UV) and delta Ct values were calculated by $\Delta$Ct = Ct [UV]−Ct [−UV].

**Library construction and deep sequencing.** Sequencing libraries were prepared from UV-ChIP, FA ChIP and control DNA fragments according to the manufacturer's protocol (NEBNext Ultra DNA Library Prep Kit). In brief, DNA fragments were end-repaired and the blunt, phosphorylated ends were treated with Klenow DNA polymerase and dATP to yield a 3′ A base overhang for ligation of adapters. After adapter ligation, DNA was PCR amplified (15 cycles). Libraries were size-selected to achieve ChIP DNA fragment lengths of 200–300 bp. The purified DNA was captured on an Illumina flow cell for cluster generation and libraries were sequenced on a HiSeq 2500 (Illumina) instrument according to manufacturer's protocol.

Total RNA was isolated from OCI-LY1 cells (RNAeasy Plus Kit, Qiagen). RNA concentration, purity and integrity was verified using a 2100 Bioanalyzer (Agilent Technologies). Libraries were generated using the mRNA-seq sample preparation kit (Illumina). Briefly, mRNA was selected by two rounds of purification using

magnetic polydT beads and then fragmented. First strand synthesis was performed using random oligonucleotides and SupersciptIII (Invitrogen). After second strand synthesis a 200 bp paired-end library was prepared following Illumina paired-end library preparation and sequencing on a HiSeq 2500 (Illumina) instrument according to manufacturer's protocol.

Raw sequence data files are publicly available at NCBI Gene Expression Omnibus (GEO) (http://www.ncbi.nlm.nih.gov/geo) under accession no. GSE103125.

**Data processing.** Sequence data from UV-ChIP, FA ChIP and control experiments were used for further analysis. Reads were mapped onto the human genome reference (GRCh37/hg19) using Bowtie (−l 50–3 1 −n 2 −m 1 --best --strata)[46]. For all analysis, only uniquely mapped reads including a maximum of two mismatches were accepted and redundant reads with identical coordinates were filtered out. Reads were screened against the blacklist regions (collection of signal artifacts) in the human genome (https://sites.google.com/site/anshulkundaje/projects/blacklists) and overlapping reads were removed[33]. Strand cross-correlation (SCC) analysis and computation of NSC (normalized strand coefficient) and RSC (relative strand coefficient) values was performed using phantompeakqualtools (https://code.google.com/p/phantompeakqualtools) with default parameters (−s = −500:5:1500)[47]. For the Irreproducible Discovery Rate (IDR) framework, we applied peak detection by SPP (−npeak = 300,000) on replicate and pseudo-replicate (pooled and subsampled reads from replicates) data and used peak calling results ranked by signal value for consistency analysis (IDR 0.01)[47,48]. Code and detailed instructions using the IDR framework are available at https://sites.google.com/site/anshulkundaje/projects/idr. MACS2 was used for peak calling in sequence data with default parameters ($p$-value 1e$^{-2}$)[35]. For further analysis, a window of 250 bp around the peak summit (base position of maximum enrichment) was defined and peaks were ranked based on their peak calling $p$-value. For comparative studies, we used published ChIP-seq raw data (GSE29282 and GSE46663) and performed data processing as described above[30,31]. DNA sequences (FASTA, GRCh37/hg19) were generated from chromosome coordinates produced by peak detection and windowing using BEDTools[49]. De novo motif analysis was performed using MEME (-dna -mod zoops -nmotifs 5 -revcomp -minw 5 -maxw 30) on DNA sequences of windowed top ranked peaks (500)[50]. TOMTOM was used to compare identified motifs against the JASPAR database[36,51]. The position weight matrix (PWM) of the canonical BCL6 motif was mapped onto DNA sequences of windowed peaks using FIMO ($p$-value 1e$^{-4}$, $q$-value 0.1)[36,52]. The distance of canonical BCL6 motifs ($q$-value 0.1) relative to the corresponding peak summit ($x$ axis, base position of maximum enrichment, ±250 bp) was calculated and plotted over all peaks. Conservation analysis was performed using the ChIPseeqerCons module of ChIPseeqer[53]. Conservation scores centered at the peak summit were computed as the mean placental mammal conservation level (phastCons) extracted from hg19 phastCons46way.placental track (UCSC Genome Browser database)[54]. Randomly selected sequences were used as a control. For DNaseI hypersensitive site (DHS) analysis we used data (bed format) from human B-cell lymphoma cells (OCI-LY7) (GSE86713, https://www.encodeproject.org/ENCSR489NAM) and the DNaseI Hypersensitive Site Master List (125 cell types) from ENCODE/Analysis (http://hgdownload.cse.ucsc.edu/goldenpath/hg19/encodeDCC/wgEncodeAwgDnaseMasterSites) and the overlap was determined to windowed peak lists using BEDTools[33]. Highly Occupied Target (HOT) regions were used from human B cells (GM12878) and processed equally (http://encodenets.gersteinlab.org/metatracks)[38]. Data for histone marks H3K27ac (GSM763424) and H3K4me3 (GSM763420) in OCI-LY1 cells were used from Hatzi et al. (GSE29282) and processed equally[30]. Peak locations were analyzed based on human RefSeq gene annotations (GRCh37/hg19). Peaks localized ± 2 kb of the transcription start site (TSS) of annotated genes were defined as promoter peaks, peaks localized to intronic, exonic and UTR regions were defined as intragenic peaks and peaks away from intragenic and promoter regions were defined intergenic peaks. Target gene prediction based on detected binding sites was performed using BETA (http://cistrome.org/BETA) with default parameters[42]. Read density profiles (signal tracks) were generated as pileup signal files (normalized by fragment pileup in million reads) ($q$-value 5e$^{-2}$) and fold enrichment tracks were generated by normalized ChIP over input DNA pileup signal files using MACS2[35]. Profiles were visualized by the Integrative Genomics Viewer (IGV)[55].

RNA sequencing reads were trimmed by Trimmomatic (0.36) to remove low-quality sequences and reads that were shorter than 36 bases[56]. The processed paired-end RNA sequencing reads were aligned to the human genome using TopHat (2.1.1) with default parameters and the human reference genome and its annotation files were obtained from the Illumina iGenomes collection (Ensembl GRCh37) (http://cole-trapnell-lab.github.io/cufflinks/igenome_table)[57]. The mapped reads were assembled using Cufflinks (2.2.1) with default parameters to determine gene and transcript expression (FPKM, fragments per kilobase of transcript per million mapped reads)[58]. A $t$-test was performed to test for significant differences between the mean FPKM values. For differential gene expression (DGE) analysis we used RNA-seq data 48 h after BCL6 or control small interfering RNA (siRNA) transduction in OCI-LY1 cells (GSE29282)[30]. The processing of the paired-end RNA sequencing reads was performed as described and differentially expressed genes ($q$-value 0.05) were identified between control (siNT) and BCL6 knockdown (siBCL6) RNA-seq data using Cuffdiff with default

parameters[58]. Paired *t*-test was performed to test for significant differences between the mean FPKM values.

**Data availability**. Raw sequence data and processed files are publicly available at NCBI Gene Expression Omnibus (GEO) (http://www.ncbi.nlm.nih.gov/geo) under accession no. GSE103125.

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

## Acknowledgements

The project was funded in part by the Federal Ministry of Education and Research and the Free State of Thuringia, Germany. This work was supported by the Deutsche Forschungsgemeinschaft (SCHE 1909/2-1 to T.S.). We wish to thank Katerina Hatzi (Ari Melnick Lab) and the ENCODE Project Consortium for making their data publicly available. We gratefully acknowledge John Chan (City of Hope) for valuable discussions, A. Bleicher, T. Fuchs, and S. Stengel for helpful comments on the manuscript, the GATC Biotech team (Konstanz, Germany) for the sequencing service, G. Mrotzek, M. Roth, K. Volling, A. Heidrich, and P. Hennersdorf for technical and bioinformatic assistance and T. Steube for editing the manuscript.

## Author contribution

A.S., A.T., and H.P.S conceived the project; A.S. and A.T. designed and performed the experiments; A.S., T.S., and A.T. analyzed the data; A.S., T.S., A.T., and H.P.S. wrote the manuscript.

## Additional information

**Competing interests:** The authors declare no competing financial interests.

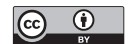

