## [Peer Review File · Nature Communications]

Reviewers' comments:

Reviewer #1 (Remarks to the Author):

In this manuscript, the authors performed ChIP-seq of BCL6 in lymphoma cell line OCI-Ly1. Instead of using conventional formaldehyde (FA) crosslinking, they treated cells with high-intensity ultraviolet (UV) laser to crosslink protein-DNA interaction hoping to catch direct binding sites of BCL6 as UV induces formation of covalent "zero-length" crosslinks between nucleotide bases and protein amino acids in immediate contact. They identified roughly 7,000 BCL6 binding sites. They found these sites are enriched in canonical BCL6 motifs. When they compared their BCL6 binding sites to previously published BCL6 binding sites captured by FA-ChIP-seq in the same cell line, they observed large discrepancies, such as low promoter binding and low overlap with DNaseI hypersensitive sites in UV-ChIP-seq. They found that UV-ChIP-seq captured some previously validated BCL6 binding sites, but the majority of the peaks were previously undetected by FA-ChIP-seq. They attributed that to whether those were direct or indirect binding of BCL6 at these genomic loci.

It is intriguing that UV crosslink revealed a largely different BCL6 binding profile in the same cell line. The authors stopped short at validating their results, fully characterizing the binding sites captured by their method (i.e. target genes, etc), detailed comparison between their dataset and FA-ChIP-seq results, and experimental evidence that could explain the difference.

Major comments:

1. The authors identified 6,777 BCL6 binding sites using UV-ChIP-seq. According to Supplementary Table 2, it appeared that the duplication rate of ChIP-seq reads was quite high in both of the biological replicates (66.7% and 72.7% respectively). In addition, the authors did not show validation of BCL6 binding peaks by another method, such as ChIP-qPCR. This raises the concern regarding the quality of this ChIP-seq experiment. Furthermore, an approach to demonstrate that those binding sites identified by UV-ChIP-seq are functional is necessary here.

2. The comparison between the UV-ChIP-seq performed here and the published BCL6 FA ChIP-seq in the same cell line revealed significant difference in the genomic distribution of BCL6 binding sites. In particular, binding sites identified by UV-ChIP-seq were largely depleted from promoters. The authors speculated that BCL6 promoter binding might be indirect. It would be nice to present evidence regarding this point. For example, what is the percentage of canonical BCL6 motif at promoters? Is that significantly lower than other genomic regions, etc? In addition, the comparison between UV and FA ChIP-seq is limited. It would be nice to show detailed comparison, for example how many peaks are overlapping between two methods, what are the overlapping target genes, etc.

3. The authors stated that UV-ChIP-seq identified 4,726 new BCL6 binding sites. However, there were no experiments to show that these were true functional BCL6 binding sites. What are the genes regulated by this newly observed BCL6 binding? Do they implicate any previously unknown BCL6 functions? Without these experiments, one can't rule out the

possibility that these newly observed binding sites are just some artifacts of this technique.

4. UV is known to induce DNA damage in cells. The authors did not show whether under their UV crosslinking condition, there was DNA damage in the cells or not. Is it possible that the discrepancy between UV and FA ChIP-seqs could be due to DNA damaged induced by UV?

Minor comments:

1. The enrichment values shown in Supplementary Figure 1 are unbelievably high (50-1500 fold of enrichment). Please provide raw values and demonstrate how are the results calculated.
2. The authors stated that they compared their BCL6 binding sites to DHSs in human GC B cells they obtained from reference 35. However, there was no human GC B cell listed among the 125 cell types profiled in that reference. Please provide detailed information of the GC B cell DHS dataset used in the manuscript.

Reviewer #2 (Remarks to the Author):

This manuscript describes the use of laser UV protein-DNA crosslinking to map the distribution of a BCL6 genome-wide. The results are impressive and the authors present numerous controls that instill confidence in the technique. In particular, Figure 1d compares the results of the BCL6 UV ChIP-seq to several negative controls that are very compelling. This technique is an important addition to the genomic approaches currently available because it should only detect proteins that are in direct contact with the DNA. In addition, the 5 second irradiation time used for the crosslinking opens up the possibility of doing kinetic analyses.

While I'm impressed with the UV ChIP-seq data, I'm very concerned about how this contrasts with the formaldehyde ChIP-seq data. In an attempt to resolve this, I discovered what could be a fundamental flaw in both protocols. In the UV crosslinking procedure, lysing the cells in NP40 only could result in regions of the chromosomes not being solubilized. As far as I can tell from the Hatzi procedure, they lysed the cells in a solution with multiple detergents: NP40, deoxycholate, and SDS. This combination of detergents will temper the unfolding properties of SDS by forming mixed micelles so I suspect the absence of BCL6 sites that are associated with compact chromatin in the Hatzi study could be due to a failure to solubilize this chromatin. Both research groups have potentially introduced bias into their ChIP-seq analyses by not solubilizing and sonicating their samples in SDS alone. The problem is probably more acute with the formaldehyde crosslinked samples.

The following are additional matters to address:

- 1) Supplementary Figure 1 provides UV dose response curves. I don't really understand what is being displayed. Since these are single sample ChIP results, it would be more informative to me to see what fraction of the input DNA is being immunoprecipitated with

the BCL6 antibody and the IgG. This is a routine way to present ChIP data and would allow readers to consider how signals and background compare to standard formaldehyde ChIP experiments.

2) Plot the peak intensities from the two UV crosslinking experiments against each other. The values should cluster along a diagonal.

3) Provide a Venn diagram to display the overlap and lack thereof for peaks detected with UV and formaldehyde.

4) Address the possibility that BCL6 motif might be particularly photoactive? It is AT rich.

5) Line 168: insert the word "both" before UV-ChIP-seq.

6) The text refers to panels 3e and 3f but the letter designations are missing from figure 3.

7) In figure 1d, the read profiles for the UV-ChIP-seq are presented as read pile-ups while the formaldehyde ChIP seq is presented as fold enrichment over input. If the point is to compare these, then the data should be presented in the same way.

8) Line 371: Are the researchers really "de-crosslinking" the samples? Moreover, it would be informative to comment on why the UV crosslinking and other damage in the DNA doesn't interfere with PCR amplification of the DNA.

Revision**Manuscript: NCOMMS-16-00843-T****Title: *High-intensity UV laser ChIP-seq for the study of protein-DNA interactions in living cells***

Dear Editor and Reviewers,

Thank you for considering our manuscript “High-intensity UV laser ChIP-seq for the study of protein-DNA interactions in living cells” for publication in Nature Communications. We greatly appreciate your thoughtful comments that helped improve our manuscript. We trust that all of your comments have been addressed accordingly in the revised manuscript. Thank you very much for your effort. In this revision, together with the original data, we present a set of additional data and analyses strongly supporting the quality of the presented UV-ChIP-seq method and our conclusions. In the following, we give a point-by-point reply to your comments (reviewer comments are *italicized*, replies are in blue):

Reviewers' comments:**Reviewer #1 (Remarks to the Author):**

In this manuscript, the authors performed ChIP-seq of BCL6 in lymphoma cell line OCI-Ly1. Instead of using conventional formaldehyde (FA) crosslinking, they treated cells with high-intensity ultraviolet (UV) laser to crosslink protein-DNA interaction hoping to catch direct binding sites of BCL6 as UV induces formation of covalent "zero-length" crosslinks between nucleotide bases and protein amino acids in immediate contact. They identified roughly 7,000 BCL6 binding sites. They found these sites are enriched in canonical BCL6 motifs. When they compared their BCL6 binding sites to previously published BCL6 binding sites captured by FA-ChIP-seq in the same cell line, they observed large discrepancies, such as low promoter binding and low overlap with DNaseI hypersensitive sites in UV-ChIP-seq. They found that UV-ChIP-seq captured some previously validated BCL6 binding sites, but the majority of the peaks were previously undetected by FA-ChIP-seq. They attributed that to whether those were direct or indirect binding of BCL6 at these genomic loci.

It is intriguing that UV crosslink revealed a largely different BCL6 binding profile in the same cell line. The authors stopped short at validating their results, fully characterizing the binding sites captured by their method (i.e. target genes, etc), detailed comparison between their dataset and FA-ChIP-seq results, and experimental evidence that could explain the difference.

We sincerely thank the reviewer for the constructive criticisms and valuable comments, which were of great help in revising our manuscript. Accordingly, we have added several new experiments and analyses to the revised manuscript especially addressing validation of the method as well as validation of the newly found target genes.

Major comments:

1. The authors identified 6,777 BCL6 binding sites using UV-ChIP-seq. According to Supplementary Table 2, it appeared that the duplication rate of ChIP-seq reads was quite high in both of the biological replicates (66.7% and 72.7% respectively).

As the reviewer points out the duplication rate of UV-ChIP-seq sequencing reads was high in both biological replicates. We think that one explanation for this observation is that the higher specificity of UV photochemical crosslinking (direct protein-DNA interactions) in comparison to FA crosslinking (protein-protein and protein-DNA interactions) leads to a lower amount of DNA starting material for the sequencing library construction. Intriguingly, other UV based techniques as the widely used HITS-Clip and iClip methods for genome-wide mapping of RNA-protein interactions produce duplication rates up to 90% (PMID 27018577).

At this stage of research we did not perform intensive optimization in order to reduce the duplication rate for example by optimizing the number of PCR cycles during the library construction step.

We would like to point out that duplicated reads are filtered out before data analysis thus our analysis and the final data are not affected by the UV-ChIP-seq sequence read duplicates. In fact, the signal-to-noise ratios in our UV-ChIP-seq data are much higher than in the FA ChIP-seq data (Fig. 1d and Supplementary Fig. 7).

In addition, the authors did not show validation of BCL6 binding peaks by another method, such as ChIP-qPCR. This raises the concern regarding the quality of this ChIP-seq experiment.

We thank the reviewer for this comment as it showed us that it might not be clearly enough written in the manuscript that we performed UV-ChIP-qPCR for validation of BCL6 binding sites. UV-ChIP-qPCR of BCL6 binding sites was performed amplifying regions with canonical BCL6 DNA motifs within the promoter region of the human *CISH*, *AFF3* and *FPGT* genes, showing identical results for UV-ChIP-qPCR and UV-ChIP-seq data (Fig. 1d and Supplementary Fig. 1). To make this clear in the manuscript, we added the following phrase at Line 82 "...within the promoter regions of the *CISH*, *AFF3* and *FPGT* genes."

The overall quality of our BCL6 UV-ChIP-seq data has been extensively tested against ENCODE quality standards. Strand cross-correlation and IDR analysis showed the high quality of the individual experiments as well as their comparability (Fig. 1e and Supplementary Fig. 3 – 5). In addition, by sequencing DNA obtained from UV input, UV IgG control and -UV control ChIP experiments, we could prove that the obtained data is highly specific (Fig. 1d und Supplementary Fig. 3 and 7).

Furthermore, an approach to demonstrate that those binding sites identified by UV-ChIP-seq are functional is necessary here.

Among the BCL6 binding sites identified by UV-ChIP-seq are many functionally validated BCL6 target

genes, e.g. *CISH*, *CCND2*, *HELB* and *SOCS2* (Fig. 1d, 3c and Supplementary Fig. 7). BCL6 bound promoters and intragenic regions that are detected by UV-ChIP-seq can be found in the Supplementary Table 5 that we have added to the revised manuscript.

To further demonstrate that BCL6 binding sites detected by UV-ChIP-seq are indeed functional, we performed RNA-seq analyses in OCI-Ly1 cells. These analyses showed that genes bound by BCL6 as detected by UV-ChIP-seq are transcriptionally inactive (Fig. 4d). Moreover they were more repressed than BCL6 binding sites found by FA ChIP-seq. This further points to functionality of the bound genes found by UV-ChIP-seq as BCL6 works as a repressive transcription factor.

In addition, we analyzed both UV- and FA ChIP-seq data regarding the occurrence of BCL6 peaks at regions with high levels of histone 3 lysine 27 acetylation (H3K27ac) and histone 3 lysine 4 trimethylation (H3K4me3) (Fig. 4b). We found, that in contrast to UV-ChIP-seq, many BCL6 binding sites detected by FA ChIP-seq occurred at such sites. As BCL6 is a transcriptional repressor, recruiting histone deacetylases, this observation further strengthens the possibility that FA treatment leads to indirect or unspecific crosslinking of BCL6 to open chromatin. In turn, the fact that UV-ChIP-seq did detect almost no binding at transcriptionally active chromatin is another hint towards proper functionality of the newly detected BCL6 binding sites.

Furthermore, we would like to point out again that in contrast to FA ChIP-seq, UV-ChIP-seq detected BCL6-DNA binding predominantly at sites containing the canonical BCL6 binding motif (66.9%, n=5,100) and that these sites were significantly conserved among species (Fig. 2).

The manuscript has been adapted as follows:

To present these new results, an additional paragraph has been added to the end of the Results Section of the manuscript named "Genes containing a BCL6 binding site identified by UV-ChIP-seq are transcriptionally inactive".

The Discussion section has been restructured to embed the results. The results are discussed between Line 238 and 253.

The Online Methods section has been updated.

An additional Figure containing graphs presenting the new data has been added (Fig. 4).

Furthermore a table containing newly identified, potential target genes has been added to the Supplementary Material (Supplementary Table 5).

2. The comparison between the UV-ChIP-seq performed here and the published BCL6 FA ChIP-seq in the same cell line revealed significant difference in the genomic distribution of BCL6 binding sites. In particular, binding sites identified by UV-ChIP-seq were largely depleted from promoters. The authors speculated that BCL6 promoter binding might be indirect. It would be nice to present evidence regarding this point. For example, what is the percentage of canonical BCL6 motif at promoters? Is that significantly lower than other genomic regions, etc?

UV-ChIP-seq revealed about five times less BCL6 binding sites in promoter regions when compared to FA ChIP-seq (4.4% versus 21.1%). Nevertheless, the number of BCL6 binding sites within promoter regions containing a canonical BCL6 motif (direct sequence-specific binding) was comparable between both ChIP-seq experiments (203 versus 220). When taking into account that promoters make up only a small part of the genome, both FA as well as UV-ChIP-seq revealed a proportionally high occurrence of BCL6 binding sites at canonical DNA motifs in promoter areas. As mentioned, FA ChIP-seq found only 14% (220 out of 1610) of promoter peaks containing the canonical BCL6 motif. Moreover, as described above, promoter and enhancer binding of BCL6 overlapped with the occurrence of the histone marks H3K4me3 and H3K27ac, which are typical for active genes which was not the case for promoter peaks found by UV-ChIP (Fig. 4b). These results can be explained with the occupation of promoter regions by huge protein complexes which are crosslinked by FA to the chromatin. For example, FA treatment crosslinks even GFP in an expression-dependent manner, a proteins that has no known bona fide interactions with DNA (PMID 24173036). Therefore, the large amount of promoter peaks found by FA ChIP-seq can be explained by FA crosslinking of BCL6 bound to proteins within these areas. Further support comes from the result that the percentage of canonical BCL6 binding motifs in FA-ChIP-seq peaks found in open chromatin regions was reduced (2-3 times) in comparison to less open regions (Fig. 3a,b; see also Fig. 4b).

We have added the following sentences to the Results and Discussion Sections:

Results Section Line 177 “In addition, the number of binding sites containing a canonical BCL6 motif outside DHSs and HOT regions was 2-3 times higher then the number of peaks inside of these regions (Figures 3a,b).”.

Correlations of detected BCL6 peaks to histone modifications were added in the new paragraph “Genes containing a BCL6 binding site identified by UV-ChIP are transcriptionally inactive“.

Results are discussed between Line 240 and 253 of the Discussion Section.

In addition, the comparison between UV and FA ChIP-seq is limited. It would be nice to show detailed comparison, for example how many peaks are overlapping between two methods, what are the overlapping target genes, etc.

We agree that it was difficult to find information about the overlaps in our manuscript. We added two Venn diagrams to the manuscript showing the overlap of detected BCL6 binding sites (Figure 2c) and the overlap of BCL6 bound promoter and intragenic regions (Fig. 4c) in UV- and FA ChIP-seq experiments. We furthermore added a table (Supplementary Table 5) of potential new BCL6 target genes.

The following sentences have been added to the Results Section:

Results Section Line 165 “317 binding sites were found by both methods (Fig. 2c).”.

Results Section Line 181 "A great proportion of promoter and intergenic regions (n=742) were found to contain one or more BCL6 binding sites detected by both UV- and FA ChIP-seq analysis (Fig 4c).".

3. The authors stated that UV-ChIP-seq identified 4,726 new BCL6 binding sites. However, there were no experiments to show that these were true functional BCL6 binding sites.

What are the genes regulated by this newly observed BCL6 binding? Do they implicate any previously unknown BCL6 functions?

As described above we added several analyses, as RNA-seq and histone modification correlations, to our manuscript showing that the new BCL6 binding sites found by UV-ChIP-seq are functional (Fig. 4b,d). All analyses of our manuscript show that the detected BCL6 binding sites are more authentic than sites found by FA ChIP-seq. In addition, a table of previously known and validated as well as new BCL6 target genes has been added to the manuscript (Supplementary Table 5).

We would like to mention here that due to a mistake in our old manuscript we corrected the number of new sequence-specific BCL6 binding sites from 4,726 to 4,808.

Without these experiments, one can't rule out the possibility that these newly observed binding sites are just some artifacts of this technique.

Since it is statistically impossible that 66% (4,808 out of 7,302) of newly observed binding sites contain the canonical BCL6 DNA sequence motif by chance, the conclusion can only be that the sites detected must be authentic. We presented numerous controls (UV Input, UV IgG control and -UV control ChIP) proving the specificity of our UV-ChIP-seq technique (Fig. 1d,e and Supplementary Fig. 3 and 7). Furthermore, many BCL6 binding sites were previously found by FA ChIP-seq and have been validated by UV-ChIP-qPCR (Fig. 2c and Supplementary Fig. 1). Moreover we could show that newly found sites are conserved amongst species and that genes in their vicinity are transcriptionally repressed (Fig. 2f and 4d). These analyses indicate functionality and therefore genuineness of the detected binding sites. One can clearly rule out that, at least the newly found sites containing a canonical BCL6 motif (n=4,808), are artifacts of the UV-ChIP-seq technique.

4. UV is known to induce DNA damage in cells. The authors did not show whether under their UV crosslinking condition, there was DNA damage in the cells or not. Is it possible that the discrepancy between UV and FA ChIP-seqs could be due to DNA damaged induced by UV?

We thank the reviewer for raising this important issue as we had conducted experiments to evaluate the impact of UV laser irradiation on DNA damage.

To estimate the effects of DNA damage by high-intensity UV laser irradiation on PCR amplification,

which is the essential step in library construction for deep sequencing, we performed qPCR analysis on cells irradiated with increasing doses of UV. We irradiated cells for 0 – 60 sec (0 – 106 J/cm²) followed by qPCR amplification of DNA fragments between 124 and 295 bp, resembling the typical length of fragments used in UV-ChIP-seq experimentation (Supplementary Fig. 2). UV laser irradiation showed minimal effects on the amplificability of the DNA fragments. Only the amplification of long fragments was slightly inhibited following irradiation for more than 30 seconds (>53 J/cm²). In our BCL6 UV-ChIP-seq experiments we irradiated the cells for only 5 seconds (8.8 J/cm² (corrected from 10 J/cm² in the previous manuscript)). Therefore, the effect of UV damage is not critical for the PCR amplification of ChIP-seq libraries in our experiments.

We added a paragraph to the Results and Online Methods section as well as a Supplementary Figure illustrating the amplification of regions following UV irradiation.

The following sentences have been added to the manuscript:

Results Line 89 "This UV dose showed minimal effects on the amplificability of DNA fragments of different length as measured by qPCR (Supplementary Fig. 2).".

Online Methods Line 421 to 426.

Minor comments:

1. *The enrichment values shown in Supplementary Figure 1 are unbelievably high (50-1500 fold of enrichment). Please provide raw values and demonstrate how are the results calculated.*

In comparison to FA, UV-ChIP produces a very low background leading to these extremely high fold enrichments. A detailed description of the calculations can be found in Online Methods section Line 403 to 421 "UV dose titration for photochemical crosslinking of BCL6". For better understanding the calculation of fold enrichment has been omitted in favor of the percent of input DNA (% Input) in the revised manuscript (Supplementary Fig. 1).

2. *The authors stated that they compared their BCL6 binding sites to DHSs in human GC B cells they obtained from reference 35. However, there was no human GC B cell listed among the 125 cell types profiled in that reference. Please provide detailed information of the GC B cell DHS dataset used in the manuscript.*

The GC B cell data is not listed among the 125 cell types which were profiled by Thurman et al. but is part of the DHS data generated by the ENCODE consortium and therefore has to be cited as indicated. For better understanding, we added the GSE number of the GC B cell DHS data (GSE32970) to the Results Section Line 144 and Online Methods Line 477.

Reviewer #2 (Remarks to the Author):

This manuscript describes the use of laser UV protein-DNA crosslinking to map the distribution of a BCL6 genome-wide. The results are impressive and the authors present numerous controls that instill confidence in the technique. In particular, Figure 1d compares the results of the BCL6 UV ChIP-seq to several negative controls that are very compelling. This technique is an important addition to the genomic approaches currently available because it should only detect proteins that are in direct contact with the DNA. In addition, the 5 second irradiation time used for the crosslinking opens up the possibility of doing kinetic analyses.

While I'm impressed with the UV ChIP-seq data, I'm very concerned about how this contrasts with the formaldehyde ChIP-seq data. In an attempt to resolve this, I discovered what could be a fundamental flaw in both protocols.

In the UV crosslinking procedure, lysing the cells in NP40 only could result in regions of the chromosomes not being solubilized. As far as I can tell from the Hatzi procedure, they lysed the cells in a solution with multiple detergents: NP40, deoxycholate, and SDS. This combination of detergents will temper the unfolding properties of SDS by forming mixed micelles so I suspect the absence of BCL6 sites that are associated with compact chromatin in the Hatzi study could be due to a failure to solubilize this chromatin.

Both research groups have potentially introduced bias into their ChIP-seq analyses by not solubilizing and sonicating their samples in SDS alone. The problem is probably more acute with the formaldehyde crosslinked samples.

We thank the reviewer for his valuable comments. The use of different lysis buffer recipes as well as sonication conditions cannot be prevented when performing the two techniques. Cell lysis as well as sonication efficiency is much lower in FA crosslinked cells (PMID 19693276). In conventional FA ChIP-seq experiments, SDS containing buffers are used to lyse highly crosslinked cells and sonicate chromatin to small fragments. In buffers that don't contain strong anionic detergents, the FA crosslinked chromatin is highly resistant to fragmentation by ultrasonic treatment.

UV irradiation at high-intensity on the other hand induces a considerable amount of single and double strand DNA breaks (PMID 1946694). DNA nicks enhance the ultrasonic shearing of DNA compared to intact non-damaged DNA (PMID 19795921). Our UV laser irradiated cells were very fragile and lysed immediately with even a low amount of SDS. The resulting cell lysate was highly viscous and could therefore not be sonicated efficiently. For those reasons cells irradiated with high-intensity UV light were lysed and sonicated in milder conditions. We adjusted the original protocol from Gilmore and Lis (0.1% Sarkosyl, 0.5% NP40 for lysis of cells irradiated by low-intensity UV light) by using only 0.5% NP40 for lysis of our high-intensity UV laser irradiated cells (PMID 637964).

Omitting SDS means also that we performed IP in non-denaturing conditions similar to different native chromatin ChIP protocols. This maintains the epitopes in their native conformation, opening the path to use non-ChIP-grade antibodies raised against native proteins or peptides as it is well known that a lot of antibodies do not work in FA ChIP experimentation (PMID 19275939).

We added the following sentence to the Results and Discussion Section:

Results Line 75 “Following irradiation, cells were lysed using non-denaturing buffers and chromatin was fragmented by mild sonication.”.

Discussion Line 222 “High-intensity UV laser irradiated cells can be lysed using non-denaturing buffers. This maintains epitopes in their native conformation, opening the path to use non-ChIP-grade antibodies raised against native proteins or peptides.”.

The following are additional matters to address:

1) Supplementary Figure 1 provides UV dose response curves. I don't really understand what is being displayed. Since these are single sample ChIP results, it would be more informative to me to see what fraction of the input DNA is being immunoprecipitated with the BCL6 antibody and the IgG. This is a routine way to present ChIP data and would allow readers to consider how signals and background compare to standard formaldehyde ChIP experiments.

For better understanding we reanalyzed our UV-ChIP-qPCR data and present the fraction of the input DNA (% Input) being immunoprecipitated by the BCL6 antibody and the UV IgG control (Supplementary Fig. 1). A detailed description of the calculations can be found in the Online Methods section “UV dose titration for photochemical crosslinking of BCL6”.

2) Plot the peak intensities from the two UV crosslinking experiments against each other. The values should cluster along a diagonal.

As recommended we have plotted the peak intensities (signal value) of our BCL6 UV-ChIP-seq replicate experiments. A figure has been added to the Supplementary Material (Supplementary Fig. 5a). In addition we would like to point out that the IDR analysis compared the UV-ChIP-seq replicates and showed a high reproducibility and quality of the individual experiments. Representative UV-ChIP-seq replicate profiles are shown in Figure 1d and Supplementary Figure 7.

3) Provide a Venn diagram to display the overlap and lack thereof for peaks detected with UV and formaldehyde.

We added two Venn diagrams showing the overlap of detected BCL6 peaks (Fig. 2c) and the overlap of bound genes (promoter and intragenic regions) (Fig. 4c) in both UV- and FA ChIP-seq experiments to our manuscript.

4) *Address the possibility that BCL6 motif might be particularly photoactive? It is AT rich.*

In early *in vitro* experiments using UV photochemical crosslinking it was demonstrated that DNA binding proteins were efficiently crosslinked to thymidines but also the other nucleotide bases participate in the crosslinking reaction. For example, Angelov et al. showed that NFκB, although having many thymidines within the DNA sequence motif, is crosslinked via cytosines to dsDNA by high-intensity UV laser irradiation (PMID 12870843). Another example is RNA polymerase II which is 100 fold more efficiently crosslinked to DNA *in vivo* by UV irradiation compared to transcription factors, although promoters are usually not AT rich (PMID 7958848). This is also one of the reasons why the UV dose must be optimized for each DNA binding factor under investigation. We don't know at this time which nucleotide bases participate in the BCL6-DNA photochemical crosslinking reaction.

5) *Line 168: insert the word "both" before UV-ChIP-seq.*

We have added the word "both" before UV-ChIP-seq in Line 173.

6) *The text refers to panels 3e and 3f but the letter designations are missing from figure 3.*

We corrected our manuscript accordingly.

7) *In figure 1d, the read profiles for the UV-ChIP-seq are presented as read pile-ups while the formaldehyde ChIP seq is presented as fold enrichment over input. If the point is to compare these, then the data should be presented in the same way.*

We agree and now present the FA ChIP-seq data in Figure 1d as normalized pileup signal tracks (FA ChIP and FA Input).

8) *Line 371: Are the researchers really "de-crosslinking" the samples?*

We agree that we cannot be sure that we really "de-crosslink" our samples. Protein-DNA photochemical crosslinks were shown to be acid labile, nevertheless we possibly only elute DNA-protein-antibody complexes from the beads in acidic conditions (PMID 3949776). We therefore removed the term "de-crosslinking".

Moreover, it would be informative to comment on why the UV crosslinking and other damage in the DNA doesn't interfere with PCR amplification of the DNA.

This is a very important matter. We have added a set of experiments evaluating the effects of increasing doses of UV on DNA damage. As Reviewer #1 did ask the same question, please see response to Reviewer #1 point 4 for further details.

Reviewers' comments:

Reviewer #1 (Remarks to the Author):

In this revised manuscript the authors report a UV cross-linking ChIP method as an alternative to FA crosslinking. Given the importance of ChIP based methods in exploring transcriptional mechanisms any additional method that can provide significantly new information is of great interest. Herein the authors performed UV-ChIP on the transcription factor bcl-6 and compare it to publicly available Fa ChIP-seq performed in the same cell line.

Concerns about the methods and conclusions were provided with the initial submission and the authors have provided their responses, which partially address the original comments.

The major concern is that the authors have not performed a functional assay to assess whether the UV identified genes are repressed by Bcl-6 (as previously noted). Just showing correlations with basal gene expression is not necessarily proof of causality between bcl-6 binding and actions.

Also of note is the lack of cross-validation as noted earlier. The authors provide a few UV-PCR but do not cross validate their results. This is especially important given the heterogeneity and variability of lymphoma cell lines. The authors use ChIP-seq performed by a different group years ago as comparison. It is not clear to me what differences are due to cell line variance, different antibody lots, and other technical considerations. The proper way to compare methods is to perform them side by side with the same reagents, same library prep methods, etc.

Along these lines I was concerned by the DNase experiments. First of all the authors used normal B cells as comparison, which are profoundly different than lymphoma cell lines. Second, while it would be very interesting if a TF would mainly bind to inaccessible sites it seems quite unlikely this would be the case given the literature on this particular factor. This underlines the need for cross validation and functional assays. Demonstration that a factor like this mainly functions in sites that are not accessible would be of interest and I for one would certainly be very enthusiastic about such findings if they were convincing.

Relevant to this discussion is that the authors find UV Bcl-6 targets to be more repressed – however this may simply be due to enrichment of inaccessible chromatin. It drives home the question: are these sites of functional Bcl-6 repression?

Reviewer #2 (Remarks to the Author):

The implications of this study are quite profound given the wide-spread use of formaldehyde based ChIP. The authors have adequately addressed my previous concerns. There are a few minor points that the authors may choose to address in this revision.

- 1) line 62:change to accurate and precise
- 2) line 94: the absence of BCL6 peaks in the control lanes indicates that non covalent interactions are being disrupted. To refute any concerns that the non denaturing conditions used to lyse the cells might be a problem, the authors could note that such interactions are probably disrupted by the wash conditions.
- 3) I find it intriguing that there is far more overlap in locations (Figure 4c) than there are in BCL6 peaks (Figure 2C). Does the overlap in locations exceed what is expected by chance and does this suggest that sequence specific BCL6 interactions detected by UV could be directing the nearby BCL6 interactions detected with formaldehyde? Also, I would like to see the Venn diagrams drawn so that the sizes of the circles and overlapping regions are proportional to the numbers.
- 4) line 206 states that the BCL6 associated genes are inactive, yet there are clearly genes in each of the categories in Figure 4d that are active. The statement by the authors seems too strong. Also, define what the horizontal black line is in each box plot - is it the median value?

Revision

Manuscript: NCOMMS-16-00843-B

Title: *High-intensity UV laser ChIP-seq for the study of protein-DNA interactions in living cells*

Dear Editor and Reviewers,

We would like to express our gratitude for considering our manuscript “High-intensity UV laser ChIP-seq for the study of protein-DNA interactions in living cells” for publication in Nature Communications, and would like to thank you for your constructive criticisms and valuable comments. We addressed the raised questions to the best of our knowledge. In the revised manuscript we included new experiments and data analysis further underpinning our presented UV-ChIP-seq data and conclusions. The manuscript has been adapted accordingly in all sections and changes are marked in blue in the revised manuscript. In the following, we give a point-by-point reply to your comments (reviewer comments are *italicized*, replies are in blue):

Reviewers' comments:

Reviewer #1 (Remarks to the Author):

In this revised manuscript the authors report a UV cross-linking ChIP method as an alternative to FA crosslinking. Given the importance of ChIP based methods in exploring transcriptional mechanisms any additional method that can provide significantly new information is of great interest. Herein the authors performed UV-ChIP on the transcription factor bcl-6 and compare it to publicly available Fa ChIP-seq performed in the same cell line. Concerns about the methods and conclusions were provided with the initial submission and the authors have provided their responses, which partially address the original comments. The major concern is that the authors have not performed a functional assay to assess whether the UV identified genes are repressed by Bcl-6 (as previously noted). Just showing correlations with basal gene expression is not necessarily proof of causality between bcl-6 binding and actions.

We would like to thank the reviewer for his valuable suggestions and for taking the time reviewing our revised manuscript again. We agree that functional assays evaluating not only UV-, but also FA ChIP-seq identified BCL6 binding sites would be desirable. To proof causality between BCL6 binding and biological effects an analysis of each single binding site by e.g. mutation analyses or by massively parallel reporter assays (MPRAs), which are

extremely time consuming, would have to be performed. As our manuscript focuses on the first application of the ChIP-seq technique based on a high-intensity UV laser setup for generation of photochemical protein-DNA crosslinks in living mammalian cells, such experiments would neither prove nor disprove the functionality of the UV-ChIP-seq method. The technique itself does not allow to make conclusions concerning functionality of the protein per se but it allows to conclude functionality of the corresponding canonical DNA sequence motif. Furthermore, it is questionable whether the removal of a single transcription factor by abrogation of its binding necessarily leads to a biological effect. Moreover, we show that the detected BCL6 binding sites, strongly enriched in canonical binding motifs, are evolutionary conserved and do not overlap with activating chromatin marks, DHSs or HOT regions which strongly support the hypothesis of functionality.

We performed an additional analysis to support the hypothesis of functionality of the identified BCL6 binding sites by demonstrating that corresponding genes become regulated upon knockdown of the BCL6 protein in OCI-Ly1 cells. This further supports our data showing significant decreased expression of genes in vicinity of binding sites bound by BCL6. Taken together, all analyses strongly hint at a repressive functionality of BCL6 in OCI-Ly1 cells at the identified binding sites, most of which could not be detected by FA ChIP-seq.

We have added the following lines in the results section:

Line 196 “Target gene prediction based on identified BCL6 binding sites revealed 7,839 potential target genes in the vicinity of UV-ChIP-seq and 10,560 in the vicinity of FA ChIP-seq detected peaks (Supplementary Table 4). A number of 4,287 genes were predicted by both, UV- and FA ChIP-seq data analyses independent of a corresponding peak overlap (Fig. 4b). To evaluate the transcriptional activity of the predicted BCL6 target genes we analyzed their expression level by RNA-seq in OCI-Ly1 cells. This analysis revealed that genes associated with UV-ChIP-seq identified BCL6 binding sites showed significantly lower transcriptional activity when compared to the FA ChIP-seq target gene analysis (Fig. 4c).

Differential gene expression analyses following knockdown of BCL6 in OCI-Ly1 cells showed the significant differential regulation of 747 genes (q -value 0.05) associated with BCL6 binding sites identified by UV-ChIP-seq (Supplementary Fig. 14, Supplementary Table 5)³⁰. A number of 185 genes thereof were not predicted by FA ChIP-seq analyses and are likely to represent new functional BCL6 target genes in DLBCL cells (see Supplementary Table 5).”

We also added the mentioned figures and tables and adapted the materials and methods section accordingly.

Also of note is the lack of cross-validation as noted earlier. The authors provide a few UV-PCR but do not cross validate their results. This is especially important given the heterogeneity and variability of lymphoma cell lines. The authors use ChIP-seq performed by a different group years ago as comparison. It is not clear to me what differences are due to cell line variance, different antibody lots, and other technical considerations. The proper way to compare methods is to perform them side by side with the same reagents, same library prep methods, etc.

We agree that the best way to compare methods is to perform side by side experiments using the same materials and methods. Unfortunately, crosslinking of protein-DNA interactions, cell lysis and sonication conditions differ between FA and UV-ChIP experimentation. We embraced your suggestion and performed additional FA ChIP-seq experiments using the same cell line (OCI-Ly1) and antibody type (Bcl-6 (N-3), sc-858, Santa Cruz) as used for UV-ChIP-seq. To demonstrate that the OCI-Ly1 cells used are identical to the ones from Hatzi et al. (PMID 23911289) we performed an RNA-seq analysis showing highly comparable expression profiles in OCI-Ly1 cells (Pearson correlation 0.87). More than 85% of identified BCL6 binding sites found in our FA ChIP-seq experiments overlapped with the ones detected by Hatzi et al. (PMID 23911289) and Chapuy et al. (PMID 24332044) but only 21.3% of BCL6 binding sites overlapped with the ones identified by UV-ChIP-seq. Interestingly, almost all BCL6 binding sites (89.1%) identified by the BCL6 ChIP-seq experiments (FA and UV-ChIP-seq) contained canonical BCL6 motifs (Figure 2b and Supplementary Figure 9). This demonstrates that a combination of both, UV and FA ChIP-seq can lead to the detection of a greater number of genuine direct transcription factor binding sites.

Finally we would like to point out that our main ambition was to develop a ChIP-seq technique based on photochemical crosslinking by high-intensity nanosecond-pulsed UV laser irradiation which allows the accurate genome-wide mapping of sequence-specific transcription factors directly interacting with their DNA recognition elements in large genomes. Recently the ENCODE consortium published that most transcription factors show less than 20% sequence-specific DNA interactions (referred to as “indirect binders”) in the case of FA ChIP-seq experimentation (PMID 22955618). Studying genome-wide BCL6-DNA interactions by UV-ChIP-seq, we prove predominant direct sequence-specific DNA binding. This finding might also be true for other sequence specific factors.

The manuscript has been adapted accordingly in all sections. All figures, except figure 1, have been adapted. Text changes are marked in blue in the revised manuscript.

Along these lines I was concerned by the DNase experiments. First of all the authors used normal B cells as comparison, which are profoundly different than lymphoma cell lines.

We agree with the reviewers concerns and therefore re-analyzed our data using a recently published DNaseI hypersensitive sites (DHSs) dataset from human OCI-Ly7 cells (GSE86713) for correlation of the identified BCL6 binding sites. OCI-Ly7 cells are of the same lymphoma GCB subtype and are therefore comparable with OCI-Ly1 cells of which no DHS dataset is available (PMID 26229090). Interestingly, correlation of the UV-ChIP-seq identified binding sites to the DHS master list from ENCODE comprising DHSs from 127 human cell types (2.9 million DHSs) still showed around 50% of BCL6 binding sites (53.1%, n=4,794) do not overlap a known DHS.

The corresponding sentences have been changed as follows:

Line 131 “In order to determine whether BCL6 binding occurs preferentially in accessible or inaccessible chromatin regions, binding sites were correlated to DNaseI hypersensitive sites (DHSs) in human lymphoma cells (GSE86713). We found that UV-ChIP-seq detected the majority of BCL6 binding sites in DNaseI insensitive regions (96.6% of peaks non-overlap DHSs, n=8,723) (Fig. 3a). Interestingly, correlation of binding sites to the DHS master list from ENCODE (DHSs detected in 127 cell types, 2.9 mil DHSs) revealed 53.1% (n=4,794) of identified BCL6 peaks did not overlap a known DHS (Supplementary Fig. 8).”

Second, while it would be very interesting if a TF would mainly bind to inaccessible sites it seems quite unlikely this would be the case given the literature on this particular factor. This underlines the need for cross validation and functional assays. Demonstration that a factor like this mainly functions in sites that are not accessible would be of interest and I for one would certainly be very enthusiastic about such findings if they were convincing.

Salt fractionation by Philip W. Tucker and colleagues (PMID 16704730) proved that BCL6 is extracted in the high salt fraction from nuclease digested chromatin of isolated nuclei showing that it is mainly bound to heterochromatin. Furthermore, Albagli et al. 2000 (PMID 11046151) showed by immunofluorescence that BCL6 localizes to perinucleolar heterochromatin during the S-phase. In addition, ChIP-seq data analysis of the ENCODE project found that transcriptional repressors generally do not overlap with DNaseI hypersensitive sites (DHSs) (Thurman et al. 2012, Supplementary Figure 7, PMID 22955617).

The above studies indicate that beside its function as a transcriptional repressor, BCL6 might also be involved in the regulation of nuclear organization, which might be an important function of the binding sites within heterochromatin identified by UV-ChIP-seq.

Relevant to this discussion is that the authors find UV Bcl-6 targets to be more repressed ? however this may simply be due to enrichment of inaccessible chromatin. It drives home the question: are these sites of functional Bcl-6 repression?

We agree that the repression of UV-ChIP-seq identified potential BCL6 target genes is likely due to their locations in inaccessible chromatin. Why should our method enrich inaccessible chromatin? In contrast to FA, UV photochemical crosslinked chromatin can be very uniformly sheared. Therefore, said chromatin fraction is most likely depleted in conventional FA ChIP-seq experiments.

As mentioned above, BCL6 binding sites identified by UV-ChIP-seq in gene-poor, late replicating heterochromatin may play a role in nuclear organization. For example, Zullo et al. (PMID 22726435) demonstrated that the BTB/POZ domain containing transcriptional repressor Zbtb7b can induce silencing by anchoring chromatin to the nuclear lamina. In addition, Harr et al. (PMID 25559185) found that motifs for BTB/POZ domain containing transcriptional repressors are highly enriched in these anchoring regions. One can speculate that BCL6, which is also a member of this family of transcriptional repressors has similar functions.

Reviewer #2 (Remarks to the Author):

The implications of this study are quite profound given the wide-spread use of formaldehyde based ChIP. The authors have adequately addressed my previous concerns. There are a few minor points that the authors may choose to address in this revision.

We would like to thank the reviewer for his excellent suggestions.

1) line 62:change to accurate and precise

We changed the corresponding sentence as follows:

Line 62 “Our method enabled the accurate and precise discovery of many previously undetectable direct BCL6 binding sites, particularly in condensed, inaccessible areas of chromatin.”

2) line 94: *the absence of BCL6 peaks in the control lanes indicates that non covalent interactions are being disrupted. To refute any concerns that the non denaturing conditions used to lyse the cells might be a problem, the authors could note that such interactions are probably disrupted by the wash conditions.*

The sentence has been edited as follows:

Line 92 “To control for enrichment of non-crosslinked protein-DNA interactions we performed ChIP-seq using non-irradiated cells (-UV control ChIP). This control showed no enrichment indicating that non-covalent BCL6-DNA interactions are disrupted by the washing conditions (Fig. 1d and Supplementary Fig. 3).”

3) *I find it intriguing that there is far more overlap in locations (Figure 4c) than there are in BCL6 peaks (Figure 2C). Does the overlap in locations exceed what is expected by chance and does this suggest that sequence specific BCL6 interactions detected by UV could be directing the nearby BCL6 interactions detected with formaldehyde? Also, I would like to see the Venn diagrams drawn so that the sizes of the circles and overlapping regions are proportional to the numbers.*

The detected overlap exceeds by far the one expected by chance. We can only speculate on how sequence-specific BCL6 interactions detected by UV-ChIP-seq could be directing the nearby BCL6 interactions detected with FA ChIP-seq. It might be that in some cases where BCL6 interacts with many other proteins, indirect binding in the vicinity of the direct binding site is better detected by FA than direct binding. Also, changes in the three-dimensional

structure of DNA-binding sites might contribute to the seen effect (PMID 27581526).

The Venn diagrams have been adjusted to be proportional to the numbers.

4) line 206 states that the BCL6 associated genes are inactive, yet there are clearly genes in each of the categories in Figure 4d that are active. The statement by the authors seems too strong. Also, define what the horizontal black line is in each box plot - is it the median value?

We agree with the reviewer and changed the corresponding statements as follows:

Line 195 “UV-ChIP-seq predicted BCL6 target genes show low transcriptional activity”

Line 199 “To evaluate the transcriptional activity of the predicted BCL6 target genes we analyzed their expression level by RNA-seq in OCI-Ly1 cells. This analysis revealed that genes associated with UV-ChIP-seq identified BCL6 binding sites showed significantly lower transcriptional activity when compared to the FA ChIP-seq target gene analysis (Fig. 4c).”

The following sentence has been adapted to the figure legend of Figure 4c: „ The distribution of gene expression values (Log10 FPKM, fragments per kilobase of transcript per million mapped reads) is shown by box plots (black line represent the median value).”

REVIEWERS' COMMENTS:

Reviewer #1 (Remarks to the Author):

The authors have improved the manuscript substantially. I believe that it is plausible that this method could provide additional insight into transcription factor function, especially regarding their answers regarding possible structural functions distinct from transcription. While unproven, it is possible they could exist and this method could help future studies to explore such questions. On the other hand I fail to understand why the authors do not embrace the data they have to more conclusively validate their data. I have brought this point up each time and while they get closer - they keep on falling short.

Most importantly: with the RNAi data in hand. Why not compare and contrast the up-regulation of bcl6 targets in FA vs UV ChIP? Why not determine if inaccessible vs accesible genes have different patterns of differential expression after RNAi? Why not validate by single locus ChIP the novel Bcl6 binding sites they find? All of this is straightforward and would make their paper more compelling and interesting.

Minor point: Ly7 cells are not that similar to Ly1: they have different mutations, different biology, and I would not assume that their HS patterns are that similar. In any case I don't think is necessary to do DNase on Ly1 for this paper.